# Radiological and Functional Pulmonary Evolution in Post-COVID-19 Patients: An Observational Study

**DOI:** 10.3390/diseases11030113

**Published:** 2023-08-30

**Authors:** Virgínia Maria Cavallari Strozze Catharin, Tereza Laís Menegucci Zutin, Elen Landgraf Guiguer, Adriano Cressoni Araújo, Lucas Fornari Laurindo, Eduardo F. Baisi Chagas, Cássia Fernanda Gasparotti Zorzetto, Patrícia C. dos Santos Bueno, Manoela dos Santos Bueno, Yandra Cervelim Nunes, Vitor Cavallari Strozze Catharin, Heron Fernando Gonzaga, Sandra Maria Barbalho

**Affiliations:** 1Department of Biochemistry and Pharmacology, School of Medicine, University of Marília (UNIMAR), Avenida Hygino Muzzy Filho, 1001, Marília 17525-902, São Paulo, Brazil; virovicatharin@hotmail.com (V.M.C.S.C.); enfermagem.lais@unimar.br (T.L.M.Z.); lucasffffor@gmail.com (L.F.L.);; 2Postgraduate Program in Structural and Functional Interactions in Rehabilitation, University of Marília (UNIMAR), Avenida Hygino Muzzy Filho, 1001, Marília 17525-902, São Paulo, Brazil; 3Department of Biochemistry and Nutrition, School of Food and Technology of Marília (FATEC), Avenida Castro Alves, 62, Marília 17500-000, São Paulo, Brazil; 4Department of Biochemistry and Pharmacology, School of Medicine, Faculdade de Medicina de Marília (FAMEMA), Avenida Monte Carmelo, 800, Marília 17519-030, São Paulo, Brazil; 5Department of Animal Sciences, School of Veterinary Medicine, University of Marília (UNIMAR), Avenida Hygino Muzzy Filho, 1001, Marília 17525-902, São Paulo, Brazil; 6UNIMED Marilia-R. Bororós, 225-Sen. Salgado Filho, Marília 17502-270, São Paulo, Brazil; 7Department of Dermatology, Escola Paulista de Medicina, Federal University of São Paulo (UNIFESP), Rua dos Otonis, 861, São Paulo 04025-002, São Paulo, Brazil

**Keywords:** COVID-19, post-COVID-19, pulmonary manifestations, spirometry, chest tomography

## Abstract

COVID-19 has generated a scenario for global health with multiple systemic impairments. This retrospective study evaluated the clinical, radiological, and pulmonary functional evolution in 302 post-COVID-19 patients. Regarding post-COVID-19 pulmonary symptoms, dry cough, dyspnea, and chest pain were the most frequent. Of the associated comorbidities, asthma was more frequent (23.5%). Chest tomography (CT) initially showed a mean pulmonary involvement of 69.7%, and evaluation in the subsequent months showed improvement in the evolutionary image. With less than six months post-pathology, there was a commitment of 37.7% from six to twelve months it was 20%, and after 12 months it was 9.9%. As for most of the sample, 50.3% of the patients presented CT normalization less than six months after infection, 23% were normalized between six and twelve months, and 5.2% presented with normalized images after twelve months, with one remaining. A percentage of 17.3% maintained post-COVID-19 pulmonary residual sequelae. Regarding spirometry, less than six months after pathology, 59.3% of the patients presented regular exam results, 12.3% had their function normalized within six to twelve months, and 6.3% had normal exam results twelve months after their post-pathology evaluation. Only 3.6% of the patients still showed some alteration during this period.

## 1. Introduction

In December 2019, a case of a disease caused by the coronavirus, i.e., COVID-19, was diagnosed in Wuhan, China, leading to a worldwide pandemic of great relevance for all current humanity [1,2,3]. The epidemic disease caused by SARS-CoV-2 was called coronavirus disease-19 (COVID-19). By November 2022, Brazil had surpassed 35 million cases, associated with more than 700,000 deaths. Manifestations ranged from asymptomatic patients with mild symptoms to severe illness and death. The viral infection expanded internationally, and the WHO announced a Public Health Emergency of International Concern [1,4,5].

In approximately 80–90% of the cases, the disease may manifest with mild symptoms or asymptomatically. However, in the remaining 10%, patients may develop a significant infection, with severe dyspnea, hypoxemia, and extensive pulmonary impairment [6,7]. COVID-19 is primarily a respiratory disease, which can manifest in the upper and lower respiratory tracts. It has a progressive and gradual evolution, starting with oropharyngeal pain, rhinorrhea, nasal congestion, myalgia, arthralgia, fatigue (21–65%), fever (which becomes persistent) (81–94%), and loss of taste and smell (chemosensory disturbances) [8,9,10]. It can also evolve with a cough (65–78%), progressive dyspnea, silent hypoxia, hemoptysis, chest pain, and acute respiratory failure [9,11].

Many patients may suffer from acute post-COVID-19 syndrome, which manifests with persistent symptoms post-disease. This syndrome was defined as a condition characterized by the persistence of symptoms or complications four weeks after the onset of the disease; in some cases, it includes persistent symptoms and abnormalities after twelve weeks of the infectious condition [12,13].

Chest radiological imaging methods are fundamental, primordial, and valuable instruments for diagnosing, following up, and treating diseases [14]. Radiological diagnostics constitute an essential component for assessing the extent and severity of infection, and is a pivotal element in guiding treatment [6]. In terms of imaging, plain chest radiography can be interesting and it can be used as an initial test, but complementing it with chest computed tomography (CT) may represent a better and more detailed test for analyzing the clinical picture of COVID-19, both for the assessment of the acute phase and for post-COVID-19 syndrome [11,15,16,17,18].

In addition to the characteristic clinical scenario that may occur and the various radiographic images that should be evaluated, checking pulmonary function (spirometry) is an essential part of the follow-up appointments of these affected patients [19,20,21]. The predictive values are determined in accordance with the *Global Lung Function Initiative* [20,22]. Values below 80% of the predicted value are considered to be pathological findings (calculated based on the following variables: sex, age, height, and patient weight) [19]. The Tiffeneau index is based on values below 70% to indicate a broncho-constrictive disorder [23].

Due to the complexity of COVID-19, the responses presented in the existing literature are not exhaustive, prompting the need for further investigations to assist with the management of patients so that they have a better quality of life in the period after the onset of the disease. For these reasons, this study aimed to investigate the relationship between the clinical evolution of lung function using spirometry and computed tomography, the need for therapeutic intervention, and the type of post-COVID treatment.

## 2. Materials and Methods

### 2.1. Study Design

An observational study was performed. The data were obtained during an outpatient consultation, and access to medical records during a retroactive period of two years, until the date of diagnosis with COVID-19 was granted.

This study included 302 patients affected by COVID-19 who were treated at the Medical Specialties Clinic of the University of Marília, Sao Paulo, Brazil. All patients in the study were followed-up between March 2020 and December 2022, with the follow-up appointments occurring every six months, until the patient showed effective clinical, radiological, and functional improvement.

The eligibility requirement for participation in the study group was having a positive polymerase chain reaction (PCR) test result, with patients’ age ranging from 20 to 79 years, comprising both sexes. Those with a negative PCR test were not included, even if they exhibited compatible clinical symptoms and tested positive via an antigen test. Additionally, individuals below 20 years old or above 80 years old, pregnant women, and patients using corticosteroids were excluded from the study. The data pertaining to the included patients were gathered from records encompassing general clinical, pulmonological, functional (spirometry), and radiological assessments. These data were procured at the initial time point (onset of the disease), as well as at intervals of up to six months, six to twelve months, and beyond twelve months from the beginning of the condition. Essential identifying information like race, gender, age, and occupation, was also compiled. This study’s protocol was approved by the Research Ethics Committee of the University of Marilia, Sao Paulo, Brazil, under approval number 60359322.6.0000.5496. Patients were effectively included in the study only after they had signed the informed consent statement.

### 2.2. Statistics

The qualitative variables were described through absolute (f) and relative (%) frequency distributions. Disparities in the frequency distribution among the response categories were examined using a univariate Chi-square test. The association between qualitative variables was analyzed using Fisher’s exact test. Wald’s backward logistic regression analysis was performed to examine the effects of covariates on the CT and spirometry exams and their normalization probability. The impact of the logistic regression model was analyzed using the Omnibus test, while the model’s fitness was assessed using Cox’s R2. The significance level adopted was 5% and the data were analyzed using the SPSS software (version 24.0).

## 3. Results

The study cohort encompassed 302 patients, and the average age was 49 ± 16 years, with a minimum of 12 and a maximum of 94 years. The variation in the total absolute frequency across certain variables can be attributed to missing data. Notably, post-COVID-19 pulmonary symptoms such as dry cough, dyspnea, and chest pain were the most prevalent (data not presented). When analyzing the frequency distribution of the sex, age group, type of treatment, and morbidity concerning the year of data collection (Table 1), a significant reduction was observed in the proportion of patients who needed in-hospital treatment, intensive treatment, and intubation in the year 2022 than 2020 and 2021. Regarding comorbidities, an increase in the proportion of patients with asthma and COPD was observed in the year 2022 than 2020 and 2021.

Of the 302 patients evaluated, 191 underwent follow-up with CT and 57 with spirometry. The most significant proportion of the sample presented CT and spirometry normalization in less than six months (<6 months). However, 17.3% of the patients had no CT normalization, and 19.3% had no spirometry normalization (Table 2). The relationship between CT and spirometry normalization, and the presence of comorbidities such as diabetes, hypertension, and obesity were also analyzed. However, there were no significant differences.

Among the patients with a greater need for CT normalization time and those who did not have normalized results, a higher proportion of patients required in-hospital, intensive care unit (ICU), and orotracheal intubation (OTI) during the period of COVID treatment. Regarding post-COVID treatment, the need for inhaled medication and the combination of drugs with physiotherapy were greater among patients with a a longer CT normalization time and those who did not have normalized results (Table 3).

For the clinical evolution of spirometry, no association between priority treatment and the need for OTI during the period of COVID treatment was observed, as well as the need for post-COVID treatment. However, among patients who had a longer time to normalize their health status, a higher proportion of patients who needed ICU during the COVID treatment period was observed. Alongside that, in patients who did not have normalized spirometry results, a higher proportion of patients who did not need ICU was observed. Although no significant association was observed among patients who did not have normalized spirometry results, a higher proportion of patients with home treatment and those who did not require OTI was observed (Table 4).

A logistic regression model was constructed to analyze the effect of independent variables on the probability of CT and spirometry normalization. For the analysis, the dependent variables, CT and spirometry, were dichotomized into normalized and not normalized categories. The selection of independent variables considered the COVID treatment variables and the factors like age, sex, and morbidities. However, only the comorbidities of asthma and chronic obstructive pulmonary disease had a significant effect upon association analysis. A significant effect was observed in the initial model, but the variables explain only 18.9% of the variation in the probability of CT normalization. After excluding the independent variables that did not significantly affect the model for CT normalization, a significant effect was observed for age and the need for ICU. This explains 17.0% of the variation in the probability of normalization of CT. Increasing age and the need of ICU reduced the likelihood of CT normalization. For spirometry normalization probability, the initial model did not show a significant effect. Following the removal of independent variables that lacked a significant impact, the ultimate model demonstrated a noteworthy effect solely attributable to COPD. This particular factor contributed to 14.5% of the variance in the likelihood of spirometry normalization (Table 5).

In the logistic regression analysis, the variables of interest (detailed in Table 5) were transformed into a binary classification of “normalized” and “non-normalized” for both chest computed tomography and spirometry normalization times. The choice of independent variables for this analysis was guided by those that exhibited significant impacts on the normalization duration of chest computed tomography (indicated in Table 3) and spirometry (demonstrated in Table 4). The results displayed exclusively showcase the ultimate regression model, having excluded independent variables lacking a significant impact when using the backward method. In all models, the constant (intercept) was included. Regarding qualitative independent variables incorporated in the regression model, the numerical codes associated with the response categories were utilized to interpret the regression coefficient (B) and odds ratio.

In the logistic regression analysis, a significant effect of age (years) and ICU (0 = No; 1 = Yes) was verified on the probability of normalization of the chest tomography. These variables explain 17.0% (R2 Cox) of the variation in the likelihood of normalization of chest tomography. Increasing age and the need for ICU admission reduced the likelihood of chest CT normalization (Table 5). Regarding the possibility of spirometry normalization, only COPD morbidity (0 = No; 1 = yes) showed a significant effect. The variation in the presence of COPD explains 14.5% of the variation in spirometry normalization probability; COPD reduced the likelihood of spirometry normalization (Table 5).

Some CT photos from the representative patients can be seen in Figure 1 and Figure 2. Figure 1a shows the CT of a patient with hypertension and type 2 diabetes mellitus at early stages of the disease (August 2020), and Figure 1b shows the control tomographic image six months later. Figure 2 shows tomographic images of a patient without comorbidities at an early stage of the disease and ten months after infection. 

In our study, we did not differentiate the strains of COVID-19, since this was not our objective. However, we found that in the three years of evaluation (2020, 2021, and 2022), the predominant strain detected in 2020 and 2021 in Brazil was Delta, and in 2022 it was Omicron. Therefore, we could observe that the patients infected by the Delta strain presented more severe clinical symptoms and more clinical, radiological, and functional alterations at the post-COVID-19 stage. In 2022, the predominant strain was Omicron, and patients presented fewer clinical symptoms and radiological and functional changes in the post-COVID-19 phase.

## 4. Discussion

The involvement of COVID-19 has substantial consequences for the body, especially the respiratory system. As our study evidenced, the literature also indicates that patients who have suffered from the disease exhibit lung function patterns consistent with restrictive defects that normalize with time [6,14,24].

Similar to our results, Péterfi et al. identified that COPD, hypertension, and diabetes are the risk factors for a higher number of hospitalizations among older patients with COVID-19 [25]. Üçsular et al. [26], who carried out a retrospective study, showed that hypertension, COPD, diabetes, and cardiovascular disease in the elderly were significantly higher compared to the non-elderly. In addition, most of the elderly underwent hospital treatment in the ward and ICU. While studying the probabilities of hospitalization according to age, Watanabe et al. [27] found that older adults suffered more from hospitalizations. The literature also shows that older patients presented spirometry results of less than six months with significantly more alterations [28].

As in our study, the literature shows that, in general, most patients recover completely in clinical terms after infection with COVID-19. However, an estimated 10–15% of patients maintain symptoms for a particular time post-infection [29]. This phase is designated by clinicians as “long-term COVID-19 effects”, which includes the patients who do not recover normality within more than 2–3 weeks after infection [30,31]. In other words, post-COVID-19 syndrome is characterized by a persistent clinical picture of deterioration for about four weeks after the COVID-19 onset in a sub-chronic phase of 12 or more weeks [32,33].

Some studies compared the types of hospitalization between the different waves of the pandemic and showed that hospitalizations in the ward reduced from 22.41% among those admitted during the first wave (03/2020 to 10/2020), to 17.16% during the second (11/2020 to 06/2021), as well as the need for ICU, which reduced from 13.84% to 9.56%. A significant decrease in dyspnea was also observed (from 25.51% during the first wave to 13.13% during the second) [34].

The study by Parashar et al. [35] evaluated 255 COVID-19 survivors. Participants were classified as having mild, moderate, and severe disease, and were followed for two months. The results indicated that pulmonary function test parameters were significantly associated with disease severity, with detected obstructive and restrictive changes suggesting a mixed pattern of long-term sequelae of COVID-19. However, no significant differences were found in the peak expiratory flow and other parameters, indicating that COVID-19 is associated with a mixed pattern of spirometry results.

Diagnostic CT findings may help predict the prognosis of patients affected by COVID-19 [14,17]. The British Thoracic Society recommends radiographic evaluations for about 12 weeks after disease onset during the follow-up of patients with post-COVID-19 acute syndrome [13]. During the patient follow-up, radiological changes are frequently detected for an initially uncertain duration (the extended progression of these abnormalities still poses an unresolved question). After four months of infection, ground-glass opacities are usually observed (>40% of cases) as the most common abnormality of persistent disease [13,22]. Studies have shown the presence of signs of reticulation, with fibrous bands presenting or non-parenchymal distortion and bronchiectasis in 67% of patients who survived three months after hospital discharge and underwent orotracheal intubation and ventilatory assistance [36].

Studies show that, with time, dyspnea improves, but a subgroup maintains radiological and physiological changes [22]. Fibrotic lesions can be irreversible and lead to chronic interstitial lung disease, with a decline in lung function, worsening symptoms, and decreased quality of life, leading to early mortality [36].

Pan et al. evaluated the chest CT patterns of 209 participants, from diagnosis of the disease to one year of follow-up. Based on CT findings at 12 months, participants were categorized as having complete resolution, residual linear opacities, and multifocal cystic or reticular lesions. Full resolution occurred mainly in the first three months after the onset of symptoms, and one year after the diagnosis of COVID-19, the three CT patterns could be observed, with complete resolution being the most common [37].

In another study conducted by Corsi et al. [38], clinical status, pulmonary function tests, laboratory tests, and radiological findings were evaluated three and twelve months after discharge in patients admitted between 25 February and 2 May 2020. Twelve months after discharge, most patients had significantly improved laboratory and lung function tests. All patients with negative CT findings at three months also had a negative CT at twelve months. Among the patients who presented CT changes in three months, 2% returned to normal, 82% improved, 14% remained stably abnormal, and 2% worsened. Thus, according to the above authors, pulmonary function tests are normal in most survivors 12 months after discharge, but structural CT abnormalities may persist. Watanabe et al. [39] found that the frequency of CT changes remained high one year after infection, especially among severe/critical patients for fibrotic changes. Lerum et al. also found substantial improvement/normalization of CT over twelve months [40]. Bongiovanni et al. found that despite some abnormalities in the chest CT after twelve months, lung function impairment persists only in a minority of individuals [41]. In our results, most patients had a regular functional examination, and only a tiny percentage still had mild or moderate alterations.

Our study boasts several strengths. To begin with, the evaluation of patients included detailed information about their prior health issues and comorbidities. These factors could potentially impact the timeline of COVID-19 progression among patients, such as those with conditions like asthma, diabetes, COPD, hypertension, and smoking history, which could have compromised cardiovascular and respiratory functions. Moreover, we categorized patients based on the intensity of their treatment requirements, distinguishing between intensive medical care and home-based treatment. We also considered the necessity for orotracheal intubation and the obligations for post-COVID treatment. This characteristic lends originality and paramount relevance to our study’s findings.

Nevertheless, our study also comes with certain limitations. First, we did not distinguish between different strains of the COVID virus among the patients we included. This absence of differentiation could potentially undermine the generalizability of our study’s findings. Second, it is essential to note that our research focused solely on patients domiciled within the São Paulo state, rather than encompassing individuals from all states across Brazil. This limitation raises questions about whether regional climate variations or other environmental constraints might influence patients’ outcomes.

## 5. Conclusions

The post-COVID-19 state incurs a constellation of symptoms involving the patient’s various organs, especially those from the pulmonary tract. As for the radiological image, it was observed that 4.2% had a normal CT, 50.3% had a normalized CT in less than six months, 23% normalized between 6 and 12 months, and 5.2% had normalized images after 12 months. However, 17.3% of patients did not show CT normalization after 12 months, maintaining residual pulmonary sequelae. As for the functional status, it was found that 78% of the patients who underwent spirometry had a normal test, 15.4% had a mild change, 5.3% had a moderate change, and only 1.3% had a severe change.

Due to vaccination schedules, patients had shown changes in the presentation of the disease, unlike at the beginning of the pandemic, when the condition was intense and compromised the body systemically with significant pulmonary involvement. Our results indicate that, in most cases, there was a satisfactory clinical, radiological, and functional evolution at the post-disease stage, with most patients returning to their normal organic status within one year after infection. However, a small percentage maintained their clinical, radiological, and functional sequelae. Our results can contribute to the literature investigating the evolution of the consequences of COVID-19.

## Figures and Tables

**Figure 1 diseases-11-00113-f001:**
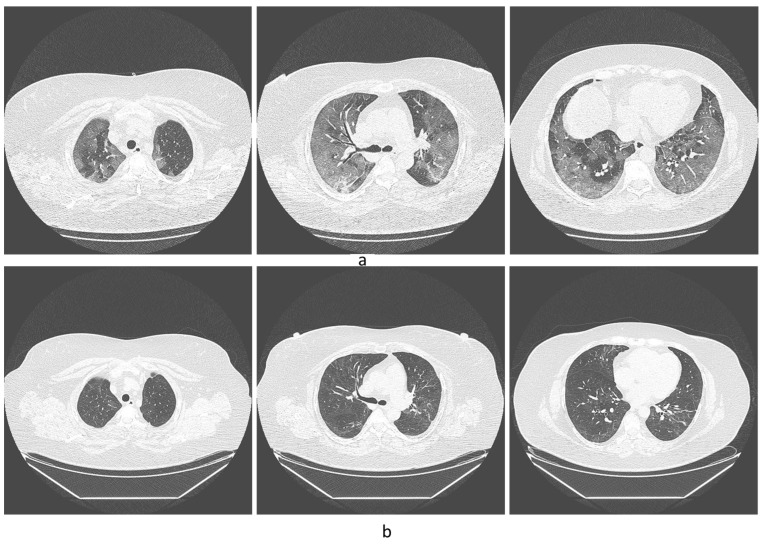
(**a**) Tomographic image of a 64-year-old patient with hypertension and type 2 diabetes mellitus at early stages of the disease, in August 2020. (**b**) Control tomographic image six months later, in February 2021.

**Figure 2 diseases-11-00113-f002:**
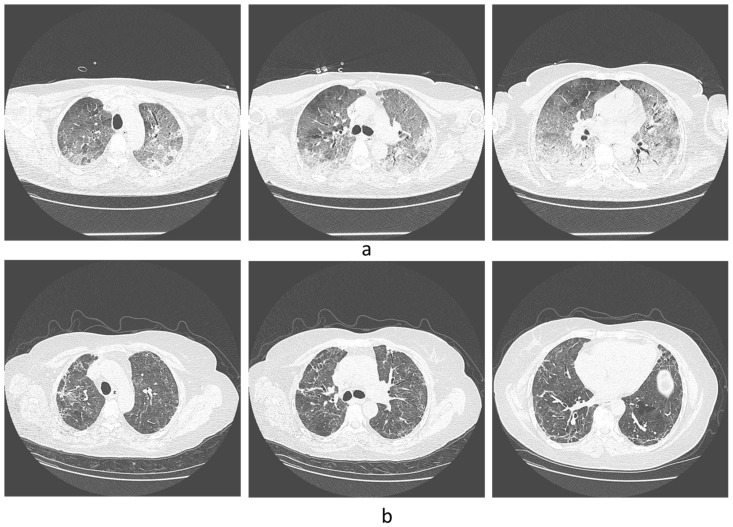
(**a**) Tomographic image of a 47-year-old patient without comorbidities at an early stage of the disease, in May 2021. (**b**) Tomographic image ten months (March 2022) after infection.

**Table 1 diseases-11-00113-t001:** Analysis of the distribution of the absolute (f) and relative (%) frequencies of data on participants, type of treatment, and presence of comorbidities concerning the collection year.

Variable	Category	Year of Collection						
		2020 (n = 41)		2021 (n = 146)		2022 (n = 115)		*p*-Value
		f	%	f	%	f	%	
Sex	Female	24	58.5	85	58.2	75	65.2	0.303
	Male	17	41.5	61	41.8	40	34.8	
Age range	<40 years	10	24.4	45	30.8	31	27.0	0.634
	40–59 years	13	31.7	74	50.7	40	34.8	
	>59 years	18	43.9	27	18.5	44	38.3	
Priority treatment	Domiciliary	28	68.3	96	65.8	110	95.7	<0.001 *
	Hospital	13	31.7	50	34.2	5	4.3	
Intensive Care Unit	No	30	73.2	129	88.4	114	99.1	<0.001 *
	Yes	11	26.8	17	11.6	1	0.9	
Orotracheal intubation	No	35	85.4	133	91.1	115	100.0	<0.001 *
	Yes	6	14.6	13	8.9	0	0.0	
Asthma	No	33	80.5	122	83.6	76	66.1	0.006 *
	Yes	8	19.5	24	16.4	39	33.9	
COPD	No	38	92.7	144	98.6	100	87.0	0.015 *
	Yes	3	7.3	2	1.4	15	13.0	
Hypertension	No	30	73.2	122	83.6	86	74.8	0.630
	Yes	11	26.8	24	16.4	29	25.2	
Diabetes	No	38	92.7	135	92.5	109	94.8	0.516
	Yes	3	7.3	11	7.5	6	5.2	
Smoke	No	32	78.0	128	87.7	95	82.6	0.904
	Yes	9	22.0	18	12.3	20	17.4	

Note: * indicates a significant difference in the proportion distribution concerning the year of collection via Fisher’s exact test, *p*-value ≤ 0.050. COPD: chronic obstructive pulmonary disease.

**Table 2 diseases-11-00113-t002:** Distribution of the absolute (f) and relative (%) frequencies of normalization time for computed tomography (CT) and spirometry.

	Category	*f*	%	*p*-Value
Normalization of chest tomography	Initial	8	4.2	<0.001 *
	<6 months	96	50.3	
	6–12 months	44	23.0	
	>12 months	10	5.2	
	No normalization	33	17.3	
	Total	191	100.0	
Normalization of spirometry	<6 months	35	61.4	<0.001 *
	6–12 months	7	12.3	
	>12 months	4	7.0	
	No normalization	11	19.3	
	Total	57	100.0	

Note: * indicates a significant difference in the proportion distribution of the response categories using the Chi-square test for a *p*-value ≤ 0.050.

**Table 3 diseases-11-00113-t003:** Analysis of the absolute (f) and relative (%) distribution of priority treatment, need for intensive care unit, need for orotracheal intubation, and post-COVID treatment with clinical evolution of computed tomography (CT).

Variable	Category		Normalization CT (n = 191)					*p*-Value
			Initial	<6 Months	6–12 Months	>12 Months	Not Normalized	
Priority treatment	Home	f (%)	8 (100.0)	82 (85.4)	21 (47.7)	4 (40.0)	12 (36.4)	<0.001 *
	Hospital	f (%)	0 (0.0)	14 (14.6)	23 (52.3)	6 (60.0)	21 (63.6)	
Intensive Care Unit	No	f (%)	8 (100.0)	94 (97.9)	36 (81.8)	9 (90.0)	18 (54.5)	<0.001 *
	Yes	f (%)	0 (0.0)	2 (2.1)	8 (18.2)	1 (10.0)	15 (45.5)	
Orotracheal intubation	No	f (%)	8 (100.0)	95 (99.0)	39 (88.6)	9 (90.0)	22 (66.7)	<0.001 *
	Yes	f (%)	0 (0.0)	1 (1.0)	5 (11.4)	1 (10.0)	11 (33.3)	
Post-COVID-19 treatment	No	f (%)	2 (25.0)	3 (3.1)	1 (2.3)	0 (0.0)	3 (9.1)	<0.001 *
	Inhaled Medication	f (%)	4 (50.0)	60 (62.5)	25 (56.8)	6 (60.0)	7 (21.2)	
	Physiotherapy	f (%)	2 (25.0)	19 (19.8)	6 (13.6)	1 (10.0)	3 (9.1)	
	Both	f (%)	0 (0.0)	14 (14.6)	12 (27.3)	3 (30.0)	20 (60.6)	

Note: * indicates a significant difference in the distribution of proportions concerning the normalization time of chest computed tomography by Fisher’s exact test, *p*-value ≤ 0.050. The relative frequency values (%) were calculated in the column within the chest computed tomography normalization time category.

**Table 4 diseases-11-00113-t004:** Analysis of the absolute (f) and relative (%) distribution of priority treatment, need for Intensive Care Unit, need for orotracheal intubation), and post-COVID treatment with clinical evolution of spirometry.

Variable	Category		Normalization of Spirometry (n = 57)				*p*-Value
			<6 Months	6–12 Months	>12 Months	No Normalization	
Priority treatment	Home	f (%)	20 (57.1)	3 (42.9)	1 (25.0)	7 (63.6)	0.561
	Hospital	f (%)	15 (42.9)	4 (57.1)	3 (75.0)	4 (36.4)	
Intensive Care Unit	No	f (%)	31 (88.6)	5 (71.4)	1 (25.0)	9 (81.8)	0.025 *
	Yes	f (%)	4 (11.4)	2 (28.6)	3 (75.0)	2 (18.2)	
Orotracheal intubation	No	f (%)	32 (91.4)	5 (71.4)	2 (50.0)	10 (90.9)	0.065
	Yes	f (%)	3 (8.6)	2 (28.6)	2 (50.0)	1 (9.1)	
Post-COVID-19 treatment	No	f (%)	2 (5.7)	0 (0.0)	0 (0.0)	1 (9.1)	0.718
	Inhaled Medication	f (%)	16 (45.7)	4 (57.1)	3 (75.0)	6 (54.6)	
	Physiotherapy	f (%)	5 (14.3)	0 (0.0)	1 (25.0)	0 (0.0)	
	Both	f (%)	12 (34.3)	3 (42.9)	0 (0.0)	4 (36.4)	

Note: * indicates a significant difference in the distribution of proportions concerning the normalization time of spirometry by Fisher’s exact test, *p*-value ≤ 0.050. The relative frequency values (%) were calculated in the column within the spirometry normalization time category.

**Table 5 diseases-11-00113-t005:** Logistic regression analysis for the effect of independent variables on the probability of computed tomography (CT) and spirometry normalization.

Variable		B	Wald	Odds	IC95% (Odds)		Model	
Dependent	Independent		*p*-Value		LL	UL	*p*-Value	R^2^ Cox
CT normalization	Age (years)	−0.043	0.004 *	0.958	0.93	0.986	<0.001 **	0.170
	ICU	−2.413	<0.001 *	0.09	0.034	0.234		
	Constant	4.37	<0.001 *	79.015				
Spirometry normalization	COPD	−2.48	0.004 *	0.084	0.016	0.443	0.003 **	0.145
	Constant	1.969	<0.001 *	7.167				

Note: ICU (0 = No;1 = yes); COPD (0 = No;1 = yes). Regression coefficient (B); * indicates a significant effect of the independent variable by the Wald test for *p*-value ≤ 0.050; odds ratio (Odds); 95% confidence interval (95%CI); lower limit (LL); upper limit (UL); ** indicates a significant effect for the model by the Omnibus test for *p*-value ≤ 0.050. COPD: chronic obstructive pulmonary disease; ICU: intensive care unit; OTI: orotracheal intubation.

## Data Availability

Not applicable.

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
