# Peer review of "Radiological and Functional Pulmonary Evolution in Post-COVID-19 Patients: An Observational Study"

_diseases, 2023, doi:10.3390/diseases11030113_

Round 1
Reviewer 1 Report
The study ‘Radiological and Functional Pulmonary Evolution in Post- 2COVID-19 Patients: An Observational Study , reported by Virgínia Maria Cavallari Strozze Catharin1 and collegues',provides in this version an update of the literature and a presentation of the personal case studies of the functional and radiological events of the post COVID.
It would be interesting to evaluate at a distance the effects of long covid radiologically evaluated as surrogates of pulmonary hypertension, such as the dilation of the pulmonary artery trunk, the basal diameter of the right ventricle and of the inferior vena cava enlargement. Especially evaluating these parameters in subjects where symptoms and altered respiratory function tests (19%) and parenchymal radiological signs (ground glass ecc),(17%),remain.
In conclusion:
This study could be interesting if data were provided
-
References
- 1 . Pulmonary artery measurements in pulmonary hypertension: the role of computed tomography. J Thorac Imaging 2013;28(2):96–103. Crossref, Medline, Google Scholar
- 2. Increased pulmonary artery diameter on chest computed tomography can predict borderline pulmonary hypertension. Pulm Circ 2013;3(2):363–368. Crossref, Medline, Google Scholar
- 3. . Egg-and-banana sign: A novel diagnostic CT marker for pulmonary hypertension. AJR Am J Roentgenol 2018;210(6):1235–1239. Crossref, Medline,
4. Imaging of Pulmonary Hypertension in Adults: A Position Paper from the Fleischner Society Radiology 2021; 298:531–549 • https://doi.org/10.1148/radiol.2020203108 •
The study ‘Radiological and Functional Pulmonary Evolution in Post- 2COVID-19 Patients: An Observational Study , reported by Virgínia Maria Cavallari Strozze Catharin1 and collegues',provides in this version an update of the literature and a presentation of the personal case studies of the functional and radiological events of the post COVID.
It would be interesting to evaluate at a distance the effects of long covid radiologically evaluated as surrogates of pulmonary hypertension, such as the dilation of the pulmonary artery trunk, the basal diameter of the right ventricle and of the inferior vena cava enlargement. Especially evaluating these parameters in subjects where symptoms and altered respiratory function tests (19%) and parenchymal radiological signs (ground glass ecc),(17%),remain.
In conclusion:
This study could be interesting if data were provided
References
- 1 . Pulmonary artery measurements in pulmonary hypertension: the role of computed tomography. J Thorac Imaging 2013;28(2):96–103. Crossref, Medline, Google Scholar
- 2. Increased pulmonary artery diameter on chest computed tomography can predict borderline pulmonary hypertension. Pulm Circ 2013;3(2):363–368. Crossref, Medline, Google Scholar
- 3. . Egg-and-banana sign: A novel diagnostic CT marker for pulmonary hypertension. AJR Am J Roentgenol 2018;210(6):1235–1239. Crossref, Medline,
4. Imaging of Pulmonary Hypertension in Adults: A Position Paper from the Fleischner Society Radiology 2021; 298:531–549 • https://doi.org/10.1148/radiol.2020203108 •
Author Response
The study ‘Radiological and Functional Pulmonary Evolution in Post- 2COVID-19 Patients: An Observational Study, reported by Virgínia Maria Cavallari Strozze Catharin1 and collegues', provides in this version an update of the literature and a presentation of the personal case studies of the functional and radiological events of the post COVID.
It would be interesting to evaluate at a distance the effects of long covid radiologically evaluated as surrogates of pulmonary hypertension, such as the dilation of the pulmonary artery trunk, the basal diameter of the right ventricle and of the inferior vena cava enlargement. Especially evaluating these parameters in subjects where symptoms and altered respiratory function tests (19%) and parenchymal radiological signs (ground glass ecc),(17%),remain.
In conclusion:
This study could be interesting if data were provided
Response: Dear Doctor, thank you very much for your time reading and providing suggestions to our manuscript. We improved the description of the results. Please, see pages 3-13, lines 111-217. Our work aimed to demonstrate the evolution of the post Covid-19 condition in clinical, radiological, and functional terms. We did not include the evaluation of Pulmonary Hypertension and dilation of the pulmonary artery trunk because this was not our objective.
With best regards.
Reviewer 2 Report
The paper is well written and can be accepeted pending minor editing English language
The paper is well written and can be accepeted pending minor editing English language. Can be accepted for pubblication in your Journal
Author Response
Comments and Suggestions for Authors
The paper is well written and can be accepted pending minor editing English language.
Response: Dear Doctor, thank you very much for your time reading and evaluating our manuscript. We know your time is precious, and we really appreciate your opinion.
With best regards.
Reviewer 3 Report
Diseases-2550802:
Radiological and Functional Pulmonary Evolution in Post-2 COVID-19 Patients: An Observational Study.
General comments:
This retrospective study evaluated the clinical, radiological, and pulmonary functional evolution in post-COVID-19 patients. Regarding post-COVID-19 pulmonary symptoms, it was found that dry cough, dyspnea, and chest pain were the most frequent and asthma was more frequent as associated comorbidities. Chest Tomography further showed a mean pulmonary involvement is about 70%, being decreased to less than 10% after 12 months but 4% still showed some alteration. This is an interesting study regarding lung involvement and the clinical changes. However, some issues are still needed to be clarified.
Specific comments:
1. If the coronavirus variants, such as delta and omicron, can be differentiated in this study, it is worth to inform the readers to consider the disease severity of COVID-19.
2. Regarding the patients’ history, the past and/active malignancies, diabetes, and the details of medication should be addressed.
3. No laboratory data including blood cell counts, inflammatory reaction, liver and renal functions, blood glucose, and electrolytes are shown in this study. There could be some interrelationships between the respiratory and laboratory findings and their changes.
4. Some CT photos from the representative patients would be helpful for the wide range of readers to understand the present data.
5. How about the changes of oxygen saturation accompanying with CT alteration?
Author Response
General comments:
This retrospective study evaluated the clinical, radiological, and pulmonary functional evolution in post-COVID-19 patients. Regarding post-COVID-19 pulmonary symptoms, it was found that dry cough, dyspnea, and chest pain were the most frequent and asthma was more frequent as associated comorbidities. Chest Tomography further showed a mean pulmonary involvement is about 70%, being decreased to less than 10% after 12 months but 4% still showed some alteration. This is an interesting study regarding lung involvement and the clinical changes. However, some issues are still needed to be clarified.
Response: Dear Doctor, thank you very much for your suggestions. We know your time is precious, and we really appreciate your opinion.
Specific comments:
- If the coronavirus variants, such as delta and Omicron, can be differentiated in this study, it is worth to inform the readers to consider the disease severity of COVID-19.
Response: Dear Doctor, thank you very much for your time reading and providing suggestions to our manuscript. In the study, we did not differentiate the strain of Covid-19 since it was not our objective. However, we found that in the three years of evaluation (2020, 2021, and 2022), we could detect that in 2020 and 2021, the predominant strain in Brazil was Delta, and in 2022 it was Omicron. Therefore, we could observe that the patients infected by the Delta strain presented more severe clinical symptoms during the occurrence of the disease and also with more clinical, radiological, and functional alterations in the post-Covid-19 stage. In 2022, the predominant strain was Omicron, and we observed less severe conditions with fewer clinical symptoms and radiological and functional changes in the post-COVID-19 phase. We included this information in the end of the Results section. Please, see lines 254-263.
- Regarding the patients’ history, the past and/active malignancies, diabetes, and the details of medication should be addressed.
Response: Thank you for this comment. During the collection of information about the patients, the presence of comorbidities such as Systemic Arterial Hypertension and Diabetes Mellitus was questioned in order to carry out. However, we did not carry out a survey on the medications used by them.
- No laboratory data including blood cell counts, inflammatory reaction, liver and renal functions, blood glucose, and electrolytes are shown in this study. There could be some interrelationships between the respiratory and laboratory findings and their changes.
Response: Dear Doctor, thank you very much for this comment. Laboratorial parameters could have correlations with respiratory findings. However, our study aimed to evaluate CT and spirometry. For these reasons, we did not include other evaluations. ,
- Some CT photos from the representative patients would be helpful for the wide range of readers to understand the present data.
Response: Dear Doctor, thank you for this comment. We agree with you, and we included some CT photos as you suggested. Please, see Figures 1, 2, and 3 at the end of the Results section. Please, see pages 7-9.
- How about the changes of oxygen saturation accompanying with CT alteration?
Response. All patients studied in the study had their oxygen saturation measured during the clinical evaluation, but this data was not included because this was not the full objective of the study.
Dear Doctor, thank you again for your valuable comments to improve the quality of our manuscript.
With best regards.
Reviewer 4 Report
The authors have conducted a study on Covid-19 patients, especially about the evolution of clinical symptoms, radiological features, and pulmonary function. They have included 302 patients that were treated at medical facility affiliated with the University of Marilia - Sao Paulo, Brazil. The authors have been able to produce some very useful information about the evolution of Covid-19. They have noted that majority of patients showed improvement and have shown the time period in which they showed improvement. They have also done a logistic regression model, however, due it's challenging to easily understand the results of the regression analysis. There are many areas that authors need to address before this manuscript can be considered for publication. The English language needs major revision. There are many grammatical errors (example: Intensive Care Unity, Treatament, Independente just to highlight the errors various tables). Also the flow of the manuscript can be improved. The introduction section is too long. In the methods section, the authors need to provide more information about the patients were selected. Were all consecutive patients included? The inclusion criteria and exclusion criteria needs to be better explained. The authors can draw an algorithm if possible to show how the patients were selected for the study. In the statistical analysis part, I do not quite understand the rationale for providing P-values for Univariate analysis for Table 1 and Table 2! I don't think it adds any value to the table. The authors may consider to re-do the analysis if possible and divide the rows into some columns (may be based on years 2020, 2021, 2022 or some other criteria) and rather do bivariate analysis and provide P-value for those. Table 3 also be improved for presentation (for example, the authors can include N= for each column at the top of the table and just mention percentages in the rows below). Same thing can be done for Table 4. The results of logistic regression analysis in Table 5 is very challenging to comprehend - the authors need to improve the language significantly where they have described about how logistric regression modeling was performed, may even want to remove some information that creates confusion. Table 5 needs major revision to make it more representable, it also has lot of information that is not necessary. For the discussion section, the authors may want to include few lines about the strengths and weaknesses of their study. Overall, the manuscript needs major revisions.
The English language needs major revision. There are many grammatical errors (example: Intensive Care Unity, Treatament, Independente just to highlight the errors various tables). Also the flow of the manuscript needs to be improved to make it easy to read. At some places, the quality of English language makes it extremely challenging to comprehend what the authors are trying to convey - this is most evident in the results section.
Author Response
The authors have conducted a study on Covid-19 patients, especially about the evolution of clinical symptoms, radiological features, and pulmonary function. They have included 302 patients that were treated at medical facility affiliated with the University of Marilia - Sao Paulo, Brazil. The authors have been able to produce some very useful information about the evolution of Covid-19. They have noted that majority of patients showed improvement and have shown the time period in which they showed improvement. They have also done a logistic regression model, however, due it's challenging to easily understand the results of the regression analysis. There are many areas that authors need to address before this manuscript can be considered for publication. The English language needs major revision.
Response. Dear doctor, thank you very much for your valuable time correcting our manuscript.
The introduction section is too long.
Response. Dear doctor, thank you for this comment. The Introduction has 511 words. Each paragraph bring information that we consider relevant to the manuscript. Due to these reasons, if you agree, we would like to keep it as it is.
The English language needs major revision. There are many grammatical errors (example: Intensive Care Unity, Treatament, Independente just to highlight the errors various tables). Also the flow of the manuscript can be improved.
Response. Dear doctor, thank you for this appointment. The manuscript was corrected by a native.
In the methods section, the authors need to provide more information about the patients were selected. Were all consecutive patients included? The inclusion criteria and exclusion criteria needs to be better explained. The authors can draw an algorithm if possible to show how the patients were selected for the study. In the statistical analysis part, I do not quite understand the rationale for providing P-values for Univariate analysis for Table 1 and Table 2! I don't think it adds any value to the table. The authors may consider to re-do the analysis if possible and divide the rows into some columns (may be based on years 2020, 2021, 2022 or some other criteria) and rather do bivariate analysis and provide P-value for those. Table 3 also be improved for presentation (for example, the authors can include N= for each column at the top of the table and just mention percentages in the rows below). Same thing can be done for Table 4. The results of logistic regression analysis in Table 5 is very challenging to comprehend - the authors need to improve the language significantly where they have described about how logistric regression modeling was performed, may even want to remove some information that creates confusion. Table 5 needs major revision to make it more representable, it also has lot of information that is not necessary. For the discussion section, the authors may want to include few lines about the strengths and weaknesses of their study. Overall, the manuscript needs major revisions.
Response. Dear doctor, thanks for these comments. We provided better information about inclusion and exclusion criteria for patient inclusion (please, see page 2, lines 93-97). Regarding tables 1 and 2, they were used to describe the main characteristics of the sample. We agree that they are descriptive and do not require the application of inferential statistics. We also agree with the suggestion of including a comparison factor, but we only include the year of collection in Table 1, because the other factors have already been explored in Tables 3 and 4 with the variables of interest to the study. Table 2 was maintained to describe the frequency distribution of the main study variables without the influence of other factors. Tables 3 and 4 were redone according to requests. For table 5, which presents the logistic regression, the text describing the results was modified, and a description of how the regression model was built was included. In addition, table 5 was redone, and the analyzes of the initial model were removed. Thus, in Table 5 only the final regression model was left. Please see all these modifications in the Results section (pages 3-13, lines 11-217)
Comments on the Quality of English Language
The English language needs major revision. There are many grammatical errors (example: Intensive Care Unity, Treatament, Independente just to highlight the errors various tables). Also the flow of the manuscript needs to be improved to make it easy to read. At some places, the quality of English language makes it extremely challenging to comprehend what the authors are trying to convey - this is most evident in the results section.
Response: Dear doctor, we asked a native to read the manuscript.
Thank you again for your time and valuable comments.
With best regards.
Round 2
Reviewer 3 Report
2nd Review:
The authors have sufficiently improved their manuscript according to the referee’s suggestion.
Author Response
A heartfelt appreciation to you, esteemed reviewer! We dedicated ample time to enhance the manuscript as per your valuable feedback.
Have a lovely weekend!
--
LUCAS FORNARI LAURINDO Faculdade de Medicina de Marília (FAMEMA) Medical Department, School of Medicine Av. Monte Carmelo, 800, Marília 17519-030, São Paulo, BrazilReviewer 4 Report
I appreciate author's sincere efforts to address the concerns raised in earlier review. Most of the concerns are appropriately addressed. If the authors can add few lines about the strengths and weaknesses of their study, it would really add to the quality of the manuscript.
Quality of English is satisfactory now. I see that at certain places 'Intensive Care Unit' is still written at 'Intensive Care Unity'. I saw this at at-least 4 places: table 1, table 3, table 4 heading and table 4 first column.
Author Response
A heartfelt appreciation to you, esteemed reviewer! We dedicated ample time to enhance the manuscript as per your valuable feedback.
We have tackled all remaining English constraints and incorporated limitations and strengths within lines 331-346.
Rest assured that we will be promptly available for any further requirements you may have concerning our manuscript!
Have a lovely weekend!
--
LUCAS FORNARI LAURINDO Faculdade de Medicina de Marília (FAMEMA) Medical Department, School of Medicine Av. Monte Carmelo, 800, Marília 17519-030, São Paulo, Brazil